# Controlled Release of Chlorogenic Acid from Polyvinyl Alcohol/Poly(γ-Glutamic Acid) Blended Electrospun Nanofiber Mats with Potential Applications in Diabetic Foot Treatment

**DOI:** 10.3390/polym13172943

**Published:** 2021-08-31

**Authors:** Isela Sandoval-Herrera, Jorge Romero-García, Antonio Ledezma-Pérez, Carmen Alvarado-Canché, Román Torres-Lubian, Arxel De-León

**Affiliations:** 1Centro de Investigación en Química Aplicada, Boulevard Enrique Reyna 140, Saltillo Coahuila 25294, Mexico; alesi0117@gmail.com (I.S.-H.); jorge.romero@ciqa.edu.mx (J.R.-G.); carmen.alvarado@ciqa.edu.mx (C.A.-C.); roman.torres@ciqa.edu.mx (R.T.-L.); 2CONACYT-Centro de Investigación en Química Aplicada, Boulevard Enrique Reyna 140, Saltillo Coahuila 25294, Mexico

**Keywords:** Chlorogenic acid, sustainable release, polyvinyl alcohol, poly(γ-glutamic acid), electrospinning

## Abstract

This work biosynthesized poly(γ-glutamic acid) (γ-PGA) produced by Bacillus licheniformis ATCC-9945a. This material was utilized to prepare electrospun nanofibers with solutions of 10% polyvinyl alcohol (PVA) (*w*/*v*) mixed with γ-PGA at 5 and 10% w/v, intended as a wound dressing for diabetic foot treatment. These solutions were loaded with chlorogenic acid (CGA), an active hypoglycemic agent. Morphological analysis showed a decrease in size of the fibers with the combination of PVA/γ-PGA compared to pure PVA nanofibers, which was attributed to the hydrogen bonding interactions between the glutaraldehyde vapors, γ-PGA, and PVA that permitted nanofiber cross-linking and allowed CGA release. The in vitro release analysis showed that the PVA membranes reached 28% delivery after the first 24 h. Notably, the nanofiber mat with PVA blended with 5% γ-PGA reached 57% delivery, and the PVA nanofiber with 10% γ-PGA reached 66% release after the same amount of time. The rate constant for the release kinetics showed that PVA with 5% γ-PGA had a higher value than that of the other samples, reaching saturation first.

## 1. Introduction

Diabetes is a chronic degenerative disease that can be present in both adults and children. Its worldwide incidence is too high, and in recent years, the risk of dying from diabetes has increased [1]. Compounding factors that can be present along with obesity are among the most important and can lead to cardiovascular diseases [2]. According to the IDF (International Diabetes Federation), there are more than 463 million worldwide with diabetes, and it is forecast that by 2045, this number will reach 700 million [3].

Health agencies consider diabetes to be an epidemiological problem that must be addressed. Consequences of diabetes can lead to diseases such as renal insufficiency, diabetic nephropathy, and high blood pressure. In the clinic, various drugs are used to treat diabetes. Although each drug has been shown to be efficacious, on occasion, side effects or even death can occur if the daily intake is not controlled [4].

Diabetic nephropathy is an ailment that originates from high blood glucose levels and is caused by a metabolic disorder that the patient presents. Diabetic foot complications, such as ulcers, are the most common cause of hospital admissions due to complications of diabetes that can lead to limb amputation [5]. Therefore, the search for new treatments and therapeutic preventative measures to control ailments associated with diabetes and diabetic foot treatment is an essential field of study that stems from an urgent need.

Chlorogenic acid (CGA) is a bioactive phenolic compound that is found in a large variety of plant species. Several laboratories have found that CGA stimulates glucose uptake by both insulin-sensitive and insulin-resistant human adipocyte cultures [6].

The above results suggest that pure CGA could help develop new strategies to treat hyperglycemia, both locally and systematically. Martinez-Jimenez et al. conducted a randomized clinical trial that demonstrated that the local application of insulin stimulates vascularization and granulation tissue formation in the wounds of diabetic patients [7]. These results strongly suggest that glycemic control in the injured area allows for more efficient healing in patients with a diabetic foot condition. Barahui et al. demonstrated in an in vitro study that a nanocomposite drug carrier made with graphene oxide and CGA for cancer treatment had significant toxicity toward some cancer cell lines but not regular cell lines [8]. Additionally, in 2005, Martinez et al. [9]. reported treating diabetes with CGA from the extracts of Mexican plants, and in 2008, Basoli et al. [10]. analyzed its effects on reducing the glucose peak in plasma, indicating a glycemic index-lowering role. These ethnopharmacological studies support CGA as a traditional medicine. Thus, CGA can help to heal injuries caused by diabetic foot ulcers because it has no toxic effects, its intake is not allergenic and its application does not cause harm [11]. Therefore, great importance has been placed on systems that permit CGA release with the final aim of mitigating the damage caused by diabetes.

Electrospinning is a versatile and effective technique for producing nanostructured biomaterial scaffolds with a large surface area and high porosity. Therefore, electrospun nanofibers have become an exciting alternative in drug delivery, healing, and tissue engineering. They have been used successfully to immobilize vitamins, enzymes, and other bioactive compounds and are attractive alternatives for the release of these compounds. They provide advantages such as a large surface area that increases the contact area between the encapsulated mixture and the release medium [12,13,14].

Moreover, in the field of pharmacology, electrospun nanofibers have generated significant interest thanks to biofunctional systems. They allow a direct interaction between polymers and drugs, and for this reason, it is of paramount importance to find a polymer that is compatible with the drug and that both materials are biocompatible [15].

The use of biodegradables polymers in release systems permits the therapeutic effect and reduces toxicity [16]. The PVA is a hydrophilic polymer that can be used in different applications like hydrogels, electrospun nanofiber mats, and biomaterials for low toxicity and biocompatibility [17]. The γ-PGA is a biodegradable and hydrophilic biopolymer with nontoxicity and good biocompatibility [18]. It is used for drug release in cancer treatment [19], gene therapy [20], antimicrobial therapy [21], diabetes therapy [22]. Other applications of γ-PGA have been developed for bone regeneration [23], 3D bioprinting with medical and surgical uses [24], cartilage regeneration [25], which can carry out for carboxylic acid (–COOH) presence, that can be easily modified for γ-PGA cross-linking and hydrogel formations.

In this work, we used CGA encapsulated in polyvinyl alcohol (PVA) and blended with γ-PGA for constructing electrospun nanofiber mats that show sustained release of an antidiabetic healing compound with potential medical applications in diabetic foot treatment. We demonstrated that CGA was successfully incorporated into PVA or PVA/γ-PGA nanofiber mats through electrospinning and released into phosphate buffer media in a sustained manner for more than 200 h. Finally, in vitro drug release profiles were obtained by analyzing the effects of incorporating γ-PGA on CGA release kinetics.

## 2. Materials and Methods

### 2.1. Materials

Polyvinyl alcohol (98% hydrolyzed; average Mw 126 kg/mol) (9002-89-5), polyvinyl alcohol (87–89% hydrolyzed; average Mw 13–23 kg/mol) (9002-89-5), chlorogenic acid (95%) (327-97-9), sodium phosphate dibasic (99%) (7558-79-4), and dialysis tubing (32 mm) (D7884) were purchased from Sigma-Aldrich Chemical Company (Burlington, MA, USA). Sodium phosphate monobasic (98%) was supplied by J.T. Baker (Phillipsburg, NJ, USA).

### 2.2. Biosynthesis of Poly (γ-glutamic acid)

The biosynthesis of γ-PGA was carried out by growing *Bacillus licheniformis* ATCC-9945a using “E” medium adjusted to pH 7, as previously described in the literature [26]. The microorganisms were incubated at 37 °C in a shaking incubator at 250 rpm for 24–48 h (New Brunswick scientific, Enfield, CT, USA). γ-PGA was purified by centrifugation, and the supernatant was washed two times with a mixture of acetone and deionized water (1:1 *v*/*v*). Then, the precipitate was collected, lyophilized, and stored in a refrigerator.

### 2.3. Preparation of PVA and PVA/γ-PGA Nanofibers by Electrospinning

The nanofiber mats were prepared by using 10% solutions of PVA (high and low molecular weights) and PVA blended with 5 or 10% γ-PGA. The different nanofibers are referred to as PVA, PVA/γ-PGA 5, and PVA/γ-PGA 10, respectively. CGA at a concentration of 0.6 mg/mL was added to each of these solutions. Before electrospinning, the solutions were loaded into a 5 mL plastic syringe. Then, the samples were injected at a flow rate of 0.2 mL/h, maintaining a syringe-collector distance of 15 cm and an applied voltage of 20 kV. Once the nanofiber mats were electrospun, they were cross-linked using glutaraldehyde [27]. Typically, the PVA and PVA/γ-PGA nanofiber mats were kept in contact with glutaraldehyde vapor for different time periods inside a desiccator. After that, the samples were withdrawn from the desiccator and heat-treated at 40 °C in an oven for 20 min. Finally, the samples were stored in zip-lock plastic bags and kept in the refrigerator until further characterization.

### 2.4. Chlorogenic Acid Release

The release of CGA from the polymeric electrospun mats was quantified by UV/Vis spectroscopy. A calibration curve was constructed using CGA standards in phosphate buffer solution (1 M at pH 4.8) at concentrations ranging from 0.5–160 μg/mL and measuring the absorbance at 324 nm.

To determine the release kinetics of CGA from the electrospun nanofibers, the samples were added to closed dialysis tubing cellulose membranes containing 80 mL of phosphate buffer and dialyzed against the same buffer solution. The different electrospun mats were incubated in a shaker at 37 °C and 250 rpm. At certain time intervals, the amount of CGA released from each system was monitored by removing 2 mL from the buffer solution outside each dialysis membrane. The amount of CGA released at that time was measured by UV-Vis spectroscopy as described posteriorly. The volume of the sample withdrawn from the outer dialysis bag solution was replaced with fresh phosphate-buffered solution at each predetermined time interval.

### 2.5. Characterization Techniques

NMR spectra were acquired on DELTA 300 MHz equipment for 1H NMR (Bruker, Billerica, MA, USA). For this analysis, 10 mg of γ-PGA was dissolved in D_2_O under an inert atmosphere. The molecular weight was determined by APC with a Waters UPLC, Agilent (Santa Clara, CA, USA) equipped with a UV detector operated at 215 nm with a silicon column and an aqueous solution of NaHPO_4_:acetonitrile at a ratio of 4:1 as the eluent. ATR Fourier transform infrared (FT-IR) analysis was carried out with a Nicolet iS5 Thermo Scientific iD7ATR (Waltham, MA, USA) ranging from 4000–600 cm^−1^ at a resolution of 16 cm^−1^ with 64 scans, using germanium as the reference standard material. SEM characterization was performed with a JEOL JSM-7041F (Akishima, Tokyo, Japan). The samples were gold-palladium sputter-coated, the size distribution and porous size was determined with ImageJ software version 1.43 (Maryland, USA). Absorption spectra were obtained with a Shimadzu 2401PC spectrophotometer (Shimadzu corporation, Kyoto, Japan).

## 3. Results

### 3.1. γ-PGA Characterization

Prior to using γ-PGA to prepare the nanofiber mats, this polymer was characterized by ^1^H NMR spectroscopy to confirm its purity. The ^1^H NMR spectrum showed peaks that corresponded to γ-PGA (Figure 1); the α-CH proton appeared at 4.2 ppm and the β-CH_2_ group had two signals at 2.16 and 1.9 ppm with the γ-CH_2_ at 2.37 ppm [28].

### 3.2. Presence of γ-PGA in the PVA Nanofibers

The presence of γ-PGA in the nanofiber mats was determined by FT-IR spectroscopy (Figure 2). In the case of PVA, the presence of the O-H stretching vibration at 3298 cm^–1^ and the methyl groups at 2940 and 2898 cm^−1^ overlap with the peaks corresponding to the aliphatic portion of γ-PGA at 3073 and 2932 cm^−1^ (Appendix A) [28]. An increase in amount of added γ-PGA resulted in the appearance of a band for amine group N–H stretching at 1590 cm^−1^ and 1640 cm^−1^ for PVA/γ-PGA 5 and PVA/γ-PGA 10, respectively. The band at 3307 cm^−1^ was attributed to the OH stretching mode. The –CH bending vibration appeared at 1436 cm^–1^, the C–N stretching vibration or overlap to C–OH for PVA that occurred at 1248 cm^−1^ and C–O stretching was observed at 1132 cm^−1^, which corresponds to the C–O–C bond that confirmed the cross-linking process. The peaks from the reaction of the OH group of PVA and the –C– groups of glutaraldehyde overlaps with those from γ-PGA and the acetyl group (C–O) that is characteristic of this polymer at 1090 cm^−1^ [29]. The molecular weight of γ-PGA obtained through the biosynthetic method with the *Bacillus licheniformis* ATTC9945a strain was Mw = 243,023 g/mol (Appendix A).

### 3.3. Morphology

Highly hydrolyzed PVA is difficult to use in electrospinning due to its high surface tension. We previously overcame this problem by blending partially hydrolyzed low molecular weight PVA with highly hydrolyzed larger molecular weight PVA. We think that the former polymer provides better mechanical properties and stability in the solvent. In contrast, the latter polymer improves the electrospinning ability of the blended polymer solution [12].

The morphology and diameter of the electrospun PVA nanofibers were analyzed before and after the cross-linking and CGA encapsulation processes (Appendix A). SEM morphological characterization of the nanofibers showed that using a 10% solution of PVA can easily form fibers at sizes in the nanometer range. The obtained nanofibers had an average diameter of 277 nm and they were flawless and smooth, verifying that the electrospinning conditions were optimal. Once the nanofiber mats were submitted to cross-linking by the glutaraldehyde vapor process, the fiber diameter increased to 298 nm. However, this increase in the fiber diameter was not apparent in comparison to the diameter without any cross-linking treatment, but preserve the particular large surface area of the nanofibers. (Figure 3).

The inclusion of γ-PGA in the PVA solution had a critical effect on the average nanofiber diameter, both before and after cross-linking. The addition of 5% γ-PGA reduced the average size diameter of the nanofibers to 163 nm, and after cross-linking, their diameter increased to 221 nm (Appendix A). Additionally, once the nanofibers were cross-linked, entanglement was observed. Both the fiber diameter increase and entanglement could be related to the absorption of the glutaraldehyde vapor inside the nanofiber structure. Nevertheless, the diameter increase was not apparent in comparison to the diameter without crosslinking. Hence, we concluded that glutaraldehyde vapor treatment is an effective method to crosslink PVA blended with other polymers.

Figure 3 shows the SEM micrograph for nanofiber PVA/γ-PGA 10. These nanofibers have an average size of 119 nm, a homogenous surface, and are defect-free. The increase in the amount of γ-PGA into the PVA solution had a notorious effect on the average diameter of the electrospun nanofibers. As the concentration of γ-PGA increased, there was a reduction in the average diameter. This reduction could be attributed to the strong hydrogen bonding between the OH groups of PVA and the NH_2_ groups of γ-PGA. However, the charge density of the polymer solution is crucial to obtain defect-free, thinner fibers [30]. γ-PGA is a natural polyelectrolyte, and giving it a high charge density. Therefore, increasing the concentration of γ-PGA in the PVA solution was expected to give thinner fibers and increase the conductivity of the polymer solution [31,32].

After cross-linking PVA/γ-PGA 10 with glutaraldehyde, an increase in the average diameter to 148 nm confirmed the bonding between polymer chains, which is likely due to the absorption of the glutaraldehyde vapors during polymer electrospinning.

### 3.4. In Vitro Release of CGA from the PVA and PVA/γ-PGA Fibers

This work studied the cumulative in vitro release of CGA from the nanofibers into phosphate buffer at pH 4.8, and the cumulative percentage of CGA released over time was plotted. As shown for many other systems, CGA typically showed biphasic release profiles, beginning with a burst release stage followed by a sustained release phase over time (Figure 4). The burst phase could be due to the release of free CGA molecules and CGA incorporated into the network nanofibers through noninclusion interactions. Moreover, the release of CGA from the nanofibers occurs more slowly through dissociation and related diffusion processes, resulting in the sustained release phase. Figure 4 shows that after one hour, the release of CGA from the PVA, PVA/γ-PGA 5, and PVA/γ-PGA 10 electrospun mats reached 6, 28, and 66%, respectively.

Notably, the maximum amount of CGA diffused into the buffer solutions in the three systems after 72 h were 36, 65, and 82%, respectively. The presence of γ-PGA had a positive effect by facilitating CGA release from the polymer-spun mats. This influence can be explained by the existence of negative charges that produce a repulsive force, which at pH 4.8, are located on the carboxylic acid groups of both γ-PGA and CGA [32]. A second possibility could be strong hydrogen bonding interactions among the inter and intrachain OH groups of PVA that limit the diffusion of small molecules such as CGA. Alternatively, γ-PGA is a linear natural polyelectrolyte that acts as a porogenic material, which likely forms tiny micrometric pores by increasing the surface area to facilitate CGA release from the fiber mats (Appendix A) [33,34].

By another hand, the inclusion of γ-PGA in the PVA solution had a critical effect on the average porosity. The addition of 5 and 10% γ-PGA increases the average porosity size of the meshes to 230 and 250 nm, respectively. In contrast, PVA meshes have an average porosity size of 180 nm, which corroborates the release behavior of CGA, which increases the porous size with γ-PGA presence, enhancing the release.

Drug release kinetics and the mechanisms are fundamental to describing the main properties and characteristics of a carrier system. Several kinetic models have been applied to study drug release from different systems. Usually, the release process of drug molecules from electrospun fibers can be described by zero-order, pseudo-first-order, or pseudo-second-order kinetic equations [35,36].

Herein, the in vitro release of CGA from the electrospun PVA and PVA/γ-PGA nanofiber mats fit to a pseudo-first-order kinetics model [37].
(1)$$\ln\left(q_{e}-q_{t}\right)=\ln q_{e}-k_{1}t\;\;\mathrm{Pseudo}-\mathrm{first}\;\mathrm{order}$$

Here, *k*_1_ is the rate constant obtained by the pseudo-first-order equation, *t* is the time, and *q_e_* and *q_t_* are the amount of drug released at the time of equilibrium and amount of drug released at any time, respectively. The value of *k_1_* can be obtained from the slope of the linear plot.

It was found that the above kinetic model is appropriate to describe the kinetic release process of CGA from the electrospun fiber mats. Figure 5 shows the ln(*q_e_* − *q_t_*) plots vs. *t*^0.65^ for the release of CGA at pH 4.8 and 37 °C. As seen in all the plots, a clean straight line was obtained for each sample.

Table 1 shows that the calculated values of *K*_1_ were 0.29, 0.33, and 0.11 for PVA/γ-PGA 10 and PVA/γ-PGA 5, respectively. In all three cases, the lineal correlation coefficient (R^2^) was very high, and the three values were quite similar to each other.

From the above constant velocity (*K*_1_) values, several observations were made. The lowest K_1_ value was obtained for the PVA fiber mats, indicating that CGA release took longer to reach saturation equilibrium, showing the burst effect. However, the values of *K*_1_ for the PVA/γ-PGA 10 and PVA/γ-PGA 5 fiber mats were very similar, indicating that the presence of γ-PGA influences CGA release by diffusion, and both cases, saturation was reached in less time than that of the PVA fiber mats.

Alternatively, the Peppas [38] and Weibull [39] models were applied to obtain more information regarding the type of diffusion mechanism.
(2)$$\frac{q_{t}}{q_{T}}=kt^{n}$$
(3)$$\frac{q_{t}}{q_{T}}=1-e^{-\alpha t^{\beta }}$$

Here, *q_t_* and *q_T_* represent the concentration of CGA released at any time and the total amount of this compound loaded in the fibers, respectively, *t* is the release time, *k* is a kinetic constant, n is the exponential factor that describes the mechanism of liberation (Equation (2)), *α* is a scale factor, and *β* is a form factor in Equation (3). An exponent n with a value below 0.5 in the Peppas model represents drug control release driven by diffusion, and the exponent *β* in the Weibull model was lower than 0.75 (Figure 6).

Figure 6 shows the *q_t_*/*q_T_* vs. *t* plots for the release of CGA from the fiber mats at pH 4.8 using the Peppas and Weibull models. The simulation results show that PVA, PVA/γ-PGA 10, and PVA/γ-PGA 5 fit well to the Peppas model of controlled drug release, with a fair regression coefficient (R^2^) for the PVA system, which gave the highest value of the three systems, while the R^2^ values for the PVA/γ-PGA systems were slightly worse. Additionally, the Weibull model did not fit the simulation results of the PVA system, but did show an excellent fit for the PVA/γ-PGA 10 system and a suitable fit for PVA/γ-PGA 5 system. Again, these data indicate that the drug release mechanism was predominantly diffusion-controlled from the combined PVA/γ-PGA systems, with values of the exponent n ranging from 0.19 to 0.21 [36,40], where the investigated formulation and processing variables did not alter the drug release mechanism (Table 2). The CGA release data from the pure PVA electrospun fiber mats did not fit the Weibull model, which can be attributed to the percentage of the released drug being lower than 50%. Introducing the small modification of 0.5 for 1 in this model’s equation shows an excellent fit to the experimental data (R^2^ of 99). Therefore, the Weibull model has limited applications in systems with a drug release percentage of less than 50% (Appendix A).

## 4. Discussion

γ-PGA was satisfactorily obtained by the biosynthesis process. Mixed PVA is a great candidate for a drug release system due to its biocompatibility and biodegradability and easy formation into electrospun mats.

The addition of γ-PGA to the PVA solution had a critical effect on the average nanofiber diameter both before and after cross-linking. The addition of γ-PGA reduces the average diameter of the nanofibers, and after cross-linking with glutaraldehyde, the diameter increases, which confirmed the bonding between the polymer chains.

A burst release stage was observed for drug release. The release of CGA from inside the nanofibers occurred through slower dissociation and diffusion processes; the cumulative release of CGA from PVA and the PVA/γ-PGA 5 and PVA/γ-PGA 10 electrospun mats reached 6, 28, and 66%, respectively. The maximum amount of CGA that diffused in the systems after 72 h were 36, 65, and 82%, for PVA and the PVA/γ-PGA 5 and PVA/γ-PGA 10 electrospun mats. These values were attributed to the presence of γ-PGA, which facilitated CGA release from the strong inter and intrachain bonding from the OH groups of PVA that limited small molecule diffusion. The presence of γ-PGA increased the porous size, enhancing the CGA release.

The analytical models confirmed that a diffusion effect dominated the mechanism involved in CGA release. The constant velocities (*K*_1_ values) for PVA/γ-PGA 5 and PVA/γ-PGA 10 were very similar with that of the PVA mat being lower, which could indicate that the release of CGA takes a longer time to reach saturation equilibrium by the hydrogen bonding of the intrachain OH groups of PVA.

This work successfully loaded CGA into PVA and PVA/γ-PGA electrospun fiber mats. Such systems have a significant advantage for the release of this natural antidiabetic drug, which can help to aid in injuries and accelerate wound closure.

## 5. Conclusions

Biodegradable polymer meshes were fabricated by electrospinning PVA and PVA/γ-PGA polymer solutions and then loaded with CGA to assess the effects of γ-PGA on the drug delivery system. Our results indicate that cross-linking with glutaraldehyde is an effective method to maintain the stability of these polymer mats. The presence of γ-PGA in the PVA solutions increased the in vitro drug release from electrospun fiber mats, which showed a concentration-dependent effect. However, the release profile was practically the same for all of the samples from the studied systems. The initial phase consisted of a burst release stage, which was followed by a sustained release stage. The maximum amount of CGA released was reached when 10% γ-PGA was introduced into the PVA solution, giving a cumulative release of 82% from the electrospun polymer fiber mats. γ-PGA positively affected drug release due to its partial repulsion of the negative charges of the carboxylic acid groups present in both CGA and γ-PGA at pH 7. The strong hydrogen bonding of PVA could be another explanation for this release behavior. Finally, application of the pseudo-first-order kinetic model was applicable and fit the CGA release from the polymer fiber mats. The Peppas and Weibull models indicated that the of CGA release kinetics from the PVA, PVA/γ-PGA 5, and PVA/γ-PGA 10 electrospun fiber mats were driven by a diffusion mechanism. The Weibull model does not apply to systems with a cumulative drug release of less than 50%.

## Figures and Tables

**Figure 1 polymers-13-02943-f001:**
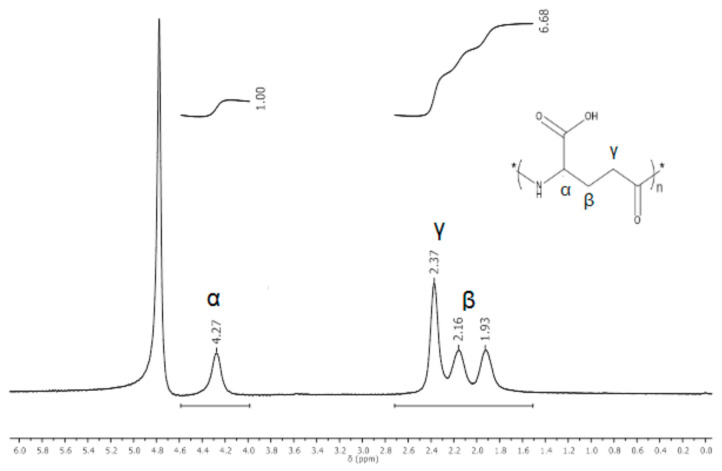
^1^H NMR spectrum of γ-PGA synthesized by *Bacillus licheniformis* ATTC9945a.

**Figure 2 polymers-13-02943-f002:**
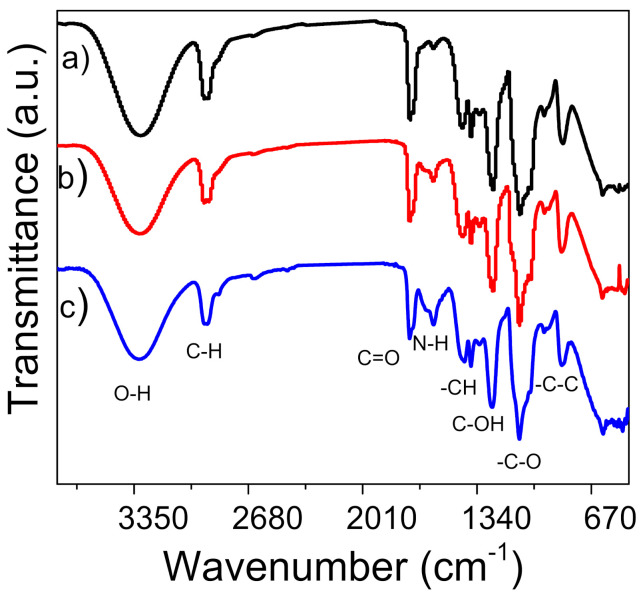
FT-IR spectra of nanofibers (a) PVA, (b) PVA/γ-PGA 5 and (c) PVA/γ-PGA 10.

**Figure 3 polymers-13-02943-f003:**
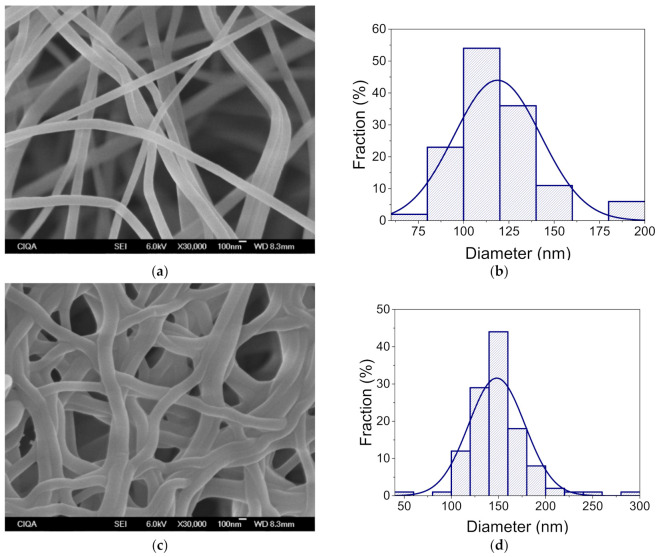
SEM micrographs of the PVA/γ-PGA 10 nanofibers, obtained by electrospinning with and without cross-linking, (**a**) and (**c**), respectively. Diameter size distribution of the nanofibers with and without cross-linking, (**b**) and (**d**), respectively.

**Figure 4 polymers-13-02943-f004:**
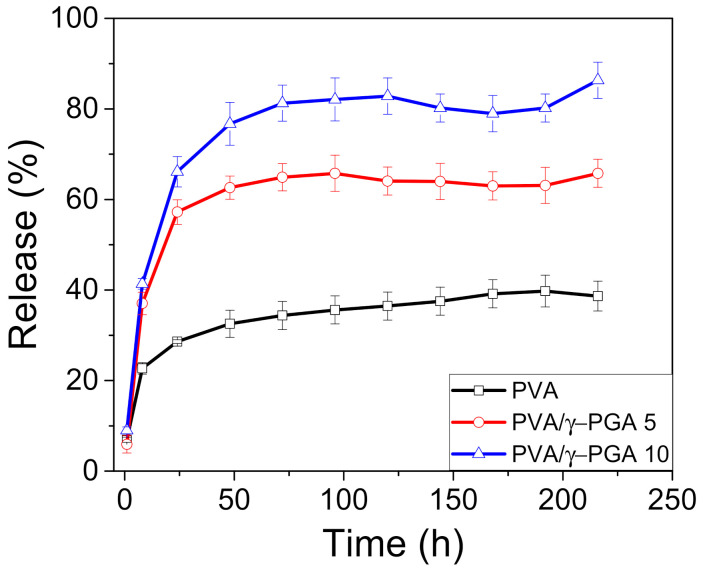
The release profiles of chlorogenic Acid from PVA, PVA/γ-PGA 5, and PVA/γ-PGA 10.

**Figure 5 polymers-13-02943-f005:**
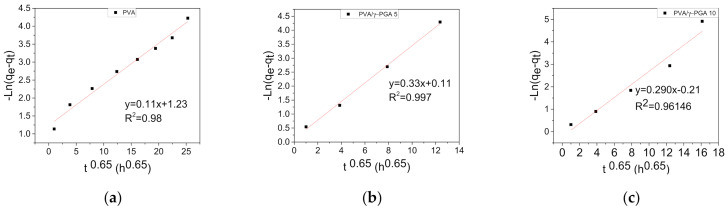
Release of CGA from electrospun fiber mats as a function of t^0.65^ at pH 4.8 and 37 °C: (**a**) PVA, (**b**) PVA/γ-PGA 5 and (**c**) PVA/γ-PGA 10.

**Figure 6 polymers-13-02943-f006:**
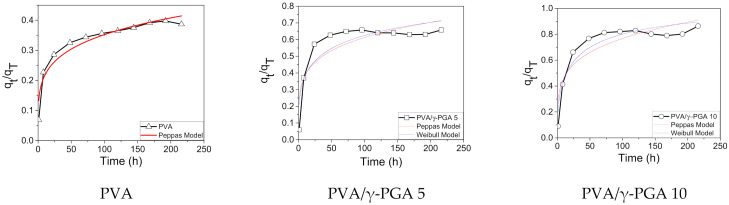
Peppas Model for PVA, PVA/γ-PGA 5, PVA/γ-PGA 10, and Weibull model for PVA/γ-PGA 5, and PVA/γ-PGA 10.

**Table 1 polymers-13-02943-t001:** Rate constants (*K*_1_) and correlation coefficients (R^2^ values) of PVA, PVA/γ-PGA 5, and PVA/γ-PGA 10.

Sample	*K* _1_ _(h_ ^−1^ _)_	R^2^ Pseudo-First-Order
PVA/γ-PGA 10	0.29	0.96
PVA/γ-PGA 5	0.33	0.99
PVA	0.11	0.98

**Table 2 polymers-13-02943-t002:** Kinetics models parameters of CGA released from loaded PVA, PVA/γ-PGA 5, and PVA/γ-PGA 10 fiber mats.

**Sample**	**Peppas Model**			**Weibull Model**		
	**η**	**K**	**R^2^**	**α**	**β**	**R^2^**
PVA/γ-PGA 10	0.21 ± 0.04	0.28 ± 0.05	0.84	0.22 ± 0.05	0.42 ± 0.05	0.92
PVA/γ-PGA 5	0.19 ± 0.04	0.24 ± 0.05	0.78	0.23 ± 0.06	0.30 ± 0.06	0.83
PVA	0.21 ± 0.02	0.13 ± 0.01	0.92	-	-	-

## Data Availability

The data presented in this study are available on request from the corresponding author.

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
