# Peer review of "Controlled Release of Chlorogenic Acid from Polyvinyl Alcohol/Poly(γ-Glutamic Acid) Blended Electrospun Nanofiber Mats with Potential Applications in Diabetic Foot Treatment"

_polymers, 2021, doi:10.3390/polym13172943_

Round 1
Reviewer 1 Report
In this article, polyvinyl alcohol/poly (γ-glutamic acid) blended electrospun nanofiber meshes have been used as a carrier of chlorogenic acid for sustained release achievement. The topic is interesting but the manuscript needs a lot of modifications to be acceptable for publication.
- I would suggest to the authors to provide a reference for the following section:
Moreover, in the field of pharmacology, electrospun nanofibers have generated significant interest thanks to biofunctional systems. They allow a direct interaction between polymers and drugs, and for this reason, it is of paramount importance to find a polymer that is compatible with the drug and that both materials are biocompatible.
- The following paragraph is not very clear:
In this work, we used CGA encapsulated in polyvinyl alcohol (PVA) and blended the PVA with γ-PGA electrospun nanofiber mats to construct a model that shows the sustained release of an antidiabetic healing compound with potential medical applications in diabetic foot treatment.
- I would suggest to the authors to clearly explain the advantages of the application of γ-PGA in blend with PVA electrospun fiber in the introduction section.
- I would ask the authors to clearly explain what is the reason for the application of crosslinker.
- I would suggest to the authors to provide the codes for all following applied materials.
Polyvinyl alcohol (98 % hydrolyzed; average Mw 126 kg/mol), polyvinyl alcohol (87–89% hydrolyzed; average Mw 13–23 kg/mol), chlorogenic acid (95 %), sodium phosphate dibasic (99 %), and dialysis tubing (32 mm) were purchased from Aldrich. Sodium phosphate monobasic (98 %) was supplied by J.T. Baker.
- I would ask the authors to explain if the concentration of polymer solution is based on weight/weight % or weight/volume? They also should explain if the ratio of γ-PGA with respect to the PVA is by weight or by volume?
- I would ask the authors to indicate which solvent(s) they have used for the preparation of different solutions.
- I would ask the authors to indicate the model of UV/Vis spectroscopy and the version of ImageJ software, that they used.
- The FT-IR spectroscopy is missing in the methods section.
- I would ask the authors to add the FTIR spectrum of pure γ-PGA in Figure 2 and write the value of each characteristic band beside the picks in the curves. These modifications would help the reviewers to better evaluate the explanation of the FTIR results provided by the authors.
- I would ask the authors to provide an explanation of why fibers diameter has significantly decreased in the presence of γ-PGA.
- The results of K1 for PVA/γ-PGA 10 and PVA/γ-PGA 5 (Table 1) are not in good agreement with their curve slopes in Figure 4, since the curve slope of PVA/γ-PGA 10 is higher than the curve slope of PVA/γ-PGA 5.
- Since the percentage of porosity of electrospun fiber meshes is one of the important parameters in drug release when we use fibers as a drug carrier system, I would suggest the authors report the porosity of different electrospun fiber meshes and discuss how this parameter affected the mechanism of release of CGA.
Author Response
Reviewer 1
In this article, polyvinyl alcohol/poly (γ-glutamic acid) blended electrospun nanofiber meshes have been used as a carrier of chlorogenic acid for sustained release achievement. The topic is interesting but the manuscript needs a lot of modifications to be acceptable for publication.
- I would suggest to the authors to provide a reference for the following section:
- Response: Thanks for your comments.
- We included the references 15
15.- Adeosun, S.O., et al., Biomaterials for Drug Delivery: Sources, Classification, Synthesis, Processing, and Applications, in Advanced Functional Materials. 2020, IntechOpen.
Moreover, in the field of pharmacology, electrospun nanofibers have generated significant interest thanks to biofunctional systems. They allow a direct interaction between polymers and drugs, and for this reason, it is of paramount importance to find a polymer that is compatible with the drug and that both materials are biocompatible [15].
- The following paragraph is not very clear:
In this work, we used CGA encapsulated in polyvinyl alcohol (PVA) and blended the PVA with γ-PGA electrospun nanofiber mats to construct a model that shows the sustained release of an antidiabetic healing compound with potential medical applications in diabetic foot treatment.
- Response: Thanks for your comments.
- We modified the sentences on lines 78-81
In this work, we used CGA encapsulated in polyvinyl alcohol (PVA) and blended with γ-PGA for constructing electrospun nanofiber mats that show sustained release of an antidiabetic healing compound with potential medical applications in diabetic foot treatment.
- I would suggest to the authors to clearly explain the advantages of the application of γ-PGA in blend with PVA electrospun fiber in the introduction section.
Response: Thanks for your comments.
We include the next sentences in the lines 78-83
The use of biodegradables polymers in release systems permits the therapeutic effect and reduces toxicity [16]. The PVA is a hydrophilic polymer that can be used in different applications like hydrogels, electrospun nanofiber mats, and biomaterials for low toxicity and biocompatibility[17]. The γ-PGA is a biodegradable and hydrophilic biopolymer with nontoxicity and good biocompatibility[18]. It is used for drug release in cancer treatment[19], gene therapy[20], antimicrobial therapy[21] and diabetes therapy[22].
- I would ask the authors to clearly explain what is the reason for the application of crosslinker.
Response: Thanks for your comments.
The PVA and γ-PGA are water solubility; after the crosslinking, we eliminated the solubility in water and kept the morphology of PVA- γ-PGA electrospun mats.
- aI would suggest to the authors to provide the codes for all following applied materials.
Polyvinyl alcohol (98 % hydrolyzed; average Mw 126 kg/mol), polyvinyl alcohol (87–89% hydrolyzed; average Mw 13–23 kg/mol), chlorogenic acid (95 %), sodium phosphate dibasic (99 %), and dialysis tubing (32 mm) were purchased from Aldrich. Sodium phosphate monobasic (98 %) was supplied by J.T. Baker.
- Response: Thanks for your comments.
We included the Cas number:
Polyvinyl alcohol (98 % hydrolyzed; average Mw 126 kg/mol ) (9002-89-5), polyvinyl alcohol (87–89% hydrolyzed; average Mw 13–23 kg/mol) (9002-89-5), chlorogenic acid (95 % ) (327-97-9), sodium phosphate dibasic (99 %) (7558-79-4), and dialysis tubing (32 mm) (D7884) and were purchased from Aldrich. Sodium phosphate monobasic (98 %) was supplied by J.T. Baker.
- I would ask the authors to explain if the concentration of polymer solution is based on weight/weight % or weight/volume? They also should explain if the ratio of γ-PGA with respect to the PVA is by weight or by volume?
- Response: Thanks for your comments.
The concentration its expressed in weight/volume for all systems.
- I would ask the authors to indicate which solvent(s) they have used for the preparation of different solutions.
- Response: Thanks for your comments.
Only use water and release; we used sodium phosphate dibasic like a buffer.
- I would ask the authors to indicate the model of UV/Vis spectroscopy and the version of ImageJ software, that they used.
- Response: Thanks for your comment
The UV-Vis is model 2401PC and brand its Shimadzu. The Image J software version is 1.43.
And included:
The samples were gold-palladium sputter-coated, and the size distribution was determined with ImageJ software version 1.43
- The FT-IR spectroscopy is missing in the methods section.
- Response: Thanks for your comment
The FT-IR spectroscopy was in the characterization techniques:
ATR Fourier transform infrared (FT-IR) analysis was carried out with a Nicolet iS5 Thermo Scientific iD7ATR ranging from 4000-600 cm-1 at a resolution of 16 cm-1 with 64 scans, using germanium as the reference standard material.
- I would ask the authors to add the FTIR spectrum of pure γ-PGA in Figure 2 and write the value of each characteristic band beside the picks in the curves. These modifications wouldhelp the reviewers to better evaluate the explanation of the FTIR results provided by the authors.
- Response: Thanks for your comment
We include the FT-IR of pure γ-PGA in supporting information Figure S1. And include the band corresponding.
Figure S1. FT-IR spectra of γ-PGA.
Figure 2. FT-IR spectra of nanofibers a) PVA, b) PVA/γ-PGA 5 and c) PVA/γ-PGA 10.
- I would ask the authors to provide an explanation of why fibers diameter has significantly decreased in the presence of γ-PGA.
- Response: Thanks for your comment
This reduction could be attributed to the strong hydrogen bonding between the OH groups of PVA and the NH2 groups of γ-PGA. However, the charge density of the polymer solution is crucial to obtain defect-free, thinner fibers[20]. γ-PGA is a natural polyelectrolyte, and giving it a high charge density. Therefore, increasing the concentration of γ-PGA in the PVA solution was expected to give thinner fibers and increase the conductivity of the polymer solution[21, 22].
- Jia, Y.-T., et al., Fabrication and characterization of poly (vinyl alcohol)/chitosan blend nanofibers produced by electrospinning method. Carbohydrate Polymers, 2007. 67(3): p. 403-409.
- McKee, M.G., et al., Solution Rheological Behavior and Electrospinning of Cationic Polyelectrolytes. Macromolecules, 2006. 39(2): p. 575-583.
- Muriel Mundo, J.L., et al., Characterization of electrostatic interactions and complex formation of ɣ-poly-glutamic acid (PGA) and ɛ-poly-l-lysine (PLL) in aqueous solutions. Food Research International, 2020. 128: p. 108781.
- The results of K1for PVA/γ-PGA 10 and PVA/γ-PGA 5 (Table 1) are not in good agreement with their curve slopes in Figure 4, since the curve slope of PVA/γ-PGA 10 is higher than the curve slope of PVA/γ-PGA 5.
Response: Thanks for your comment
The behavior was determinate in the pseudo-first-order kinetics model, the slope values were quite similar to each (0.29 to 0.33), this is an indication γ-PGA influences in the CGA release by diffusion process and the agreement is too high for this systems.
- Since the percentage of porosity of electrospun fiber meshes is one of the important parameters in drug release when we use fibers as a drug carrier system, I would suggest the authors report the porosity of different electrospun fiber meshes and discuss how this parameter affected the mechanism of release of CGA.
- Response: Thanks for your comment
- We include the next sentence in the lines 264 to 368.
- By another hand, the inclusion of γ-PGA in the PVA solution had a critical effect on the average porosity. The addition of 5 and 10% γ-PGA increases the average porosity size of the meshes to 230 and 250 nm, respectively. In contrast, PVA meshes have an average porosity size of 180 nm, which corroborates the release behavior of CGA, which increases the porous size with γ-PGA presence, enhancing the release.

Reviewer 2 Report
The manuscript is well structured and shows interesting results for the scientific community, but the quality of all the images is very bad, it must be improved for a better appreciation
Details of the experimental procedure and additional characterization are missing
What medium do they use to prepare the spinning solution?
because they didn't do NMR on the fibers as well
Because they didn't put the infrared spectrum of the synthesized polymer
kinetic results need a better explanation
Author Response
Reviewer 2
The manuscript is well structured and shows interesting results for the scientific community, but the quality of all the images is very bad, it must be improved for a better appreciation
Details of the experimental procedure and additional characterization are missing
What medium do they use to prepare the spinning solution?
- Response: Thanks for your comment
The medium used in the solution was water.
because they didn't do NMR on the fibers as well.
- Response: Thanks for your comment
Sorry for that, we don´t have NMR by solids this limit the characterization.
Because they didn't put the infrared spectrum of the synthesized polymer.
- Response: Thanks for your comment
We include the FT-IR of pure γ-PGA in supporting information Figure S1.
kinetic results need a better explanation
- Response: Thanks for your comment
We include the next sentence in the lines 361 to 363.
A burst release stage was observed for drug release. The release of CGA from inside the nanofibers occurred through slower dissociation and diffusion processes; the cumulative release of CGA from PVA and the PVA/γ-PGA 5 and PVA/γ-PGA 10 electrospun mats reached 6, 28, and 66 %, respectively. The maximum amount of CGA that diffused in the systems after 72 h were 36, 65, and 82 %, for PVA and the PVA/γ-PGA 5 and PVA/γ-PGA 10 electrospun mats. These values were attributed to the presence of γ-PGA, which facilitated CGA release from the strong inter- and intrachain bonding from the OH groups of PVA that limited small molecule diffusion.
- And include too the next sentence in the lines 264 to 368.
- By another hand, the inclusion of γ-PGA in the PVA solution had a critical effect on the average porosity. The addition of 5 and 10% γ-PGA increases the average porosity size of the meshes to 230 and 250 nm, respectively. In contrast, PVA meshes have an average porosity size of 180 nm, which corroborates the release behavior of CGA, which increases the porous size with γ-PGA presence, enhancing the release.
Round 2
Reviewer 1 Report
Thanks to the authors for providing the requested modifications.
I would suggest to the authors provide an explanation about the method that has been used for measuring the porosity of meshes in the method section.
Author Response
- Response: Thanks for your comment
- We include the sentence in the line 155.
The samples were gold-palladium sputter-coated, the size distribution and porous size was determined with ImageJ software version 1.43. Absorption spectra were obtained with a Shimadzu 2401PC spectrophotometer.

Reviewer 2 Report
I do not see any relevant change in their discussion of results, they did not take into account my comments and observations
Author Response
- Response: Thanks for your comment
Sorry, we traded to include all observations. However, it is ambiguous and modified the following sentences
This work studied the cumulative in vitro release of CGA from the nanofibers into phosphate buffer at pH 4.8, and the cumulative percentage of CGA released over time was plotted. As shown for many other systems, CGA typically showed biphasic release profiles, beginning with a burst release stage followed by a sustained release phase over time (Fig. 4). The burst phase could be due to the release of free CGA molecules and CGA incorporated into the network nanofibers through noninclusion interactions. Moreover, the release of CGA from the nanofibers occurs more slowly through dissociation and related diffusion processes, resulting in the sustained release phase. Fig. 4 shows that after one hour, the release of CGA from the PVA, PVA/γ-PGA 5, and PVA/γ-PGA 10 electrospun mats reached 6, 28, and 66%, respectively.
Notably, the maximum amount of CGA diffused into the buffer solutions in the three systems after 72 h were 36, 65, and 82 %, respectively. The presence of γ-PGA had a positive effect by facilitating CGA release from the polymer-spun mats. This influence can be explained by the existence of negative charges that produce a repulsive force, which at pH 4.8, are located on the carboxylic acid groups of both γ-PGA and CGA[29]. A second possibility could be strong hydrogen bonding interactions among the inter and intrachain OH groups of PVA that limit the diffusion of small molecules such as CGA. On the other hand, γ-PGA is a linear natural polyelectrolyte that acts as a porogenic material, which likely forms tiny micrometric pores by increasing the surface area to facilitate CGA release from the fiber mats (Table S1)[30, 31].
A burst release stage was observed for drug release. The release of CGA from inside the nanofibers occurred through slower dissociation and diffusion processes; the cumulative release of CGA from PVA and the PVA/γ-PGA 5 and PVA/γ-PGA 10 electrospun mats reached 6, 28, and 66 %, respectively. The maximum amount of CGA that diffused in the systems after 72 h were 36, 65, and 82 %, for PVA and the PVA/γ-PGA 5 and PVA/γ-PGA 10 electrospun mats. These values were attributed to the presence of γ-PGA, which facilitated CGA release from the strong inter and intrachain bonding from the OH groups of PVA that limited small molecule diffusion. The γ-PGA presence increase in the porous size, enhancing the CGA release.
The analytical models confirmed that a diffusion effect dominated the mechanism involved in CGA release. The constant velocities (K1 values) for PVA/γ-PGA 5 and PVA/γ-PGA 10 were very similar with that of the PVA mat being lower, which could indicate that the release of CGA takes a longer time to reach saturation equilibrium by the hydrogen bonding of the intrachain OH groups of PVA.
